# Behavioral and Psychiatric Symptoms in Patients with Severe Traumatic Brain Injury: A Comprehensive Overview

**DOI:** 10.3390/biomedicines11051449

**Published:** 2023-05-15

**Authors:** William Torregrossa, Loredana Raciti, Carmela Rifici, Giuseppina Rizzo, Gianfranco Raciti, Carmela Casella, Antonino Naro, Rocco Salvatore Calabrò

**Affiliations:** 1Istituti di Ricovero e Cura a Carattere Scientifico (IRCCS) Centro Neurolesi Bonino Pulejo, Via Palermo S.S. 113 C.da Casazza, 98124 Messina, Italy; william.torregrossa@irccsme.it (W.T.); carmela.rifici@irccsme.it (C.R.); gianfranco.raciti@gmail.com (G.R.);; 2Azienda Ospedaliera Universitaria (AOU) Policlinico G. Martino, Via Consolare Valeria, 1, 98124 Messina, Italyc.casella@unime.it (C.C.);

**Keywords:** psychiatric symptoms, severe traumatic brain injury, neuropsychiatric disorders, behavior

## Abstract

Traumatic brain injury (TBI) is defined as an altered brain structure or function produced by an external force. Adults surviving moderate and severe TBI often experience long-lasting neuropsychological and neuropsychiatric disorders (NPS). NPS can occur as primary psychiatric complications or could be an exacerbation of pre-existing compensated conditions. It has been shown that changes in behavior following moderate to severe TBI have a prevalence rate of 25–88%, depending on the methodology used by the different studies. Most of current literature has found that cognitive behavioral and emotional deficit following TBI occurs within the first six months whereas after 1–2 years the condition becomes stable. Identifying the risk factors for poor outcome is the first step to reduce the sequelae. Patients with TBI have an adjusted relative risk of developing any NPS several-fold higher than in the general population after six months of moderate–severe TBI. All NPS features of an individual’s life, including social, working, and familiar relationships, may be affected by the injury, with negative consequences on quality of life. This overview aims to investigate the most frequent psychiatric, behavioral, and emotional symptoms in patients suffering from TBI as to improve the clinical practice and tailor a more specific rehabilitation training.

## 1. Introduction

Approximately 150–200 per million people annually die because of traumatic brain injury (TBI) representing a core health problem [1]. 

TBI is defined as an altered brain structure or function produced by an external force. Besides the neuroradiological evidence of the acute intracranial injury, the clinical signs are neurologic impairments, loss of consciousness, amnesia for peritraumatic events, and compromised mental status. Severity of TBI has been categorized based on “the patient’s level of consciousness (assessed by the Glasgow Coma Scale [GCS]), and the duration of unconsciousness and posttraumatic amnesia (if present)” [2]. 

Neuropsychiatric symptoms such as delirium may be transient in the acute phase of the TBI, whereas mood, personality, and sleep changes, as well as psychosis and behavior impairment, are chronic symptoms that persist over time and may be seriously disabling [3,4]. Indeed, these prevent TBI patients from returning to normal activities, work, or maintaining meaningful social relationships. 

A prevalence rate of 25–88% in moderate or severe TBI has been reported [5]. 

Even though extensive research is available, the disorders onset, evolution, and risk factors for NPS after TBI is still indeterminate. 

In this overview, we sought to describe the most common neuropsychiatric and behavioral problems following TBI, indicating the prevalence, main manifestations, comorbidities, and clinical indications to better manage these important but often underestimated symptoms that may negatively affect both patients and caregivers’ quality of life. 

## 2. TBI and Risk Factors of NPS

Identifying the risk factors of poor outcome is the first step to reduce the sequelae. Recently, TBI has been considered as a disease process instead of an isolated event, with acute and chronic consequences [6]. After the injury, neuropsychiatric disturbances (NPS) can occur as primary psychiatric complications or could be an exacerbation of pre-existing compensated conditions. Lauterbach et al. have showed that NPS were higher in patients with a pre-injury history (83.2%) than those without (63.6%). Moreover, 59% showed one or more Axis I disorder before TBI and substance-use was the most common pre-injury disorder (38.5%), whereas 56.5% were diagnosed as a new diagnostic class. [7]. In all probability, premorbid factors of the psychological status, personality, contingencies, environmental reinforcements play an important role in determining the clinical picture of the “frontal personality” [8,9]. The symptoms could appear acutely or develop more gradually and insidiously influencing the grade of disability. Patients with TBI have an adjusted relative risk of developing any NPS several-fold higher than in the general population after six months of moderate-to-severe TBI [10]. A review by Babbage et al. [11] reported a rate of 39% of impairment in emotion recognition and reduced levels of empathy as well as impairment of the so-called theory of mind (ToM) [12], with detrimental effect on life satisfaction of their relatives and caregivers. [13]. 

The most common psychiatric complication associated with TBI is depression, with prevalence rates ranging between 6% and 77%, while 2–50% of TBI patients affect from generalized anxiety disorders [14]. It seems that rates of mood and anxiety disorders increase throughout the first year and that anxiety disorders emerges earlier post-injury than mood disorders [15,16,17,18]. A peak of NPS was recorded in the first year, with a subsequent significantly declined over time of anxiety disorders by 27% with each year post-injury, although mood and substance-use disorder rates persisted steady. [7].

Factors associated with depressive disorders in TBI patients include age, lifestyle, young adult, premorbid substance misuse, especially in male patients [19]. Other risk factors are lower education level [20], previous psychiatric symptoms, including anxiety, intellectual deficits and left prefrontal cortical lesions [9]. Some studies reported a frequency of NPS of 77% in males and of 71.4% in females (χ^2^ = 0.46, df = 1, *p* = 0.50). Depression in TBI individuals has been linked to decreased community integration, overall functioning, a decrease in quality of life, aggression, poor recovery, and higher rates of suicidal ideation and suicide attempts. Likewise, anxiety disorders in TBI patients are associated with poor social interpersonal functioning, decline in independent living, and acts as a positive predictive factor for the development of depression in TBI peoples. Moreover, delirium [16] and status epilepticus [21] in neurointensive care unit are very well-known risk factors for developing behavioral disorders at long term follow up. 

## 3. Neuropsychiatric Disorders in Severe TBI: Neuroanatomical Basis

The pathophysiology usually comprises volume of large brain areas, because of the high velocity impact of trauma that causes extreme motion, torsion, tensile effects, and diffuse shearing and tearing to brain structures [22]. Studies with functional magnetic resonance imaging have shown changes in task-mediated activation in the dorsolateral prefrontal cortex, ventrolateral PFC, and basal ganglia in mild TBI. In the same way, Electroencephalography has also been used to characterize and localize brain–behavior relationships with neurophysiological findings in TBI [21]. 

Pathologically, chronic traumatic encephalopathy (CTE) is characterized by an accumulation of neuronal phosphorylated tau (p-tau) in the perivascular regions and p-tau fibrils as in Alzheimer’s disease, or tauopathy [23].

TBI patients with a distinguishable structural lesion on neuroimaging have shown behavioral and emotional dyscontrol as compared to patient without brain lesions [24]. Numerous studies have documented behavioral changes in patients with very extensive unilateral-bilateral lesions of the frontal lobe [25]. A reduction in spontaneous activity and speech was the main finding, although apathy, loss of initiative, and reduction of work efficiency, as well as hyperactivity and an inability to inhibit impulsive and emotional responses, infantilism, scurrility, and fatuous euphoria were also demonstrated [25]. However, even temporal lobe structures have been commonly associated with dyscontrol syndromes, regularly in combination with the frontal lobe, as in frontotemporal degenerative illnesses [25]. Moreover, cerebellar lesions have been associated with post-TBI dyscontrol disorders [26], leading to the so-called cerebellar cognitive affective syndrome [27] The variability of symptoms and the frontal, cerebellar lesions correlation have been described in Table 1. 

### 3.1. Behavioral Dyscontrol in TBI Patients

Behavioral dyscontrol (BD) indicates a propensity to instinctively response to overall provocations. Moreover, BD have been defined as “behaviors that are noxious for the health or physical integrity of the person” [28]. It has been reported that approximately 62% of TBI patients present BD at one year from trauma, [29] described as “are not the way they used to be” and 74% at five years [30]. Brooks et al. found that the most frequent BD were irritability (64%), bad behavior (64%), drowsiness (62%), depression (57%), rapid mood change (57%), anxiety (57%), and aggression (54%) [30].

In the five years following TBI, the most common BD were irritability and lack of initiative (44%) with inappropriate social behavior (26%) [31]. 

In 2015, according to experts of the direction-finding group and based on the prevalence and incidence of BD reported in the literature, the symptoms of BD were classified into four subgroups [32]: Disruptive primary behaviors by excess (i.e., aggression, disinhibition, agitation)Disruptive primary behaviors by default (poor initiative, isolation, withdrawal)Affective disorders (anxiety, depression, somatization).Psychosis, suicide attempts and suicide.

However, classifying BD is still complex, and the Diagnostic and Statistical Manual of Mental Disorders (DSM-V) not adequately explain the complexity of neuropsychiatric symptoms of patients with TBI. In fact, DSM-V includes these symptoms as “mild” or “major” neurocognitive disorder due to TBI history diagnosis. Usually, the differential diagnosis between idiopathic and secondary symptoms is controversial. TBI patients frequently present similar symptoms of general population with NPS but without TBI history, without exactly achieving the required DSM criteria. For example, psychosis after TBI manifests with positive symptoms than negative symptoms of the general population [7]. 

Moreover, some symptoms could be due to either major depression or be consequences of frontal lobe lesion; and depression could be expressed as irritability, frustration, anger, hostility, and aggression instead of sadness [33]. Therefore, the diagnosis is usually a difficult matter.

Behavioral dyscontrol may be distinguished in (i) disinhibited type, positively associated with MRI brain volume loss, (ii) aggressive type, positively associated with more left, than right, hemispheric lesions, or (iii) combined type (if more than one of these disturbances predominate the clinical presentation) [34], characterized by global traumatic axonal injury [35,36,37].

#### 3.1.1. Disinhibition Syndromes after TBI

Disinhibition refers to socially or contextually inappropriate non-aggressive verbal, physical, and sexual acts involving a reduction or loss of inhibitions with inability to follow social or cultural behavioral norms and rules. In patients affected by TBI, this behavior usually coexists with impulsivity, as well as deficits in tolerance and social cognition [38], but its prevalence is not well established, and the frequencies range from 12% to 32% [38]. The severity of disinhibition has been associated to poor outcomes, decreased functional independence, as well as social disability. Then the symptoms should be considered during both the assessment and the training.

The assessment of disinhibition syndromes included a self-report scale, i.e., the Barratt Impulsiveness Scale (BSI). This is a 15-item short form tool providing a framework to investigate and monitor disinhibited behavior among individuals with TBI. However, the scale mainly measures impulsivity and related symptoms, and the differential diagnosis must rule out many neurologic and psychiatric disorders [39]. Then, it could not be used in the case of patients with post-traumatic self-awareness deficits. In these cases, the disinhibition subscale of the Neuropsychiatric Inventory (NPI) may provide useful data.

The treatment of posttraumatic disinhibition usually needs psychobehavioral, environmental and pharmacologic approaches. In fact, applied behavioral analysis (ABA) and treatment is a useful element of the treatment plan for disinhibited behaviors. This approach entails careful characterization of the disinhibited behaviors, their internal emotional and behavioral dyscontrol [39]. Social skills training, including individual and group interventions, with social communication skills training also may be useful. Even the telehealth group was more efficient than the in-patient group on some variables [40]. Moreover, it was demonstrated that patients with TBI had significant improvements in both goal attainment scaling (GAS) scores and qualitative evidence of enhanced social communication skills after the training with the INSIGHT program, a client-centered contextualized approach that promotes positive identity construction with consideration of efficacy of treatment [41,42]. On the other hand, there is no specific therapy for disinhibition. Pharmacological treatment of impulsivity and sexual disinhibition is variable and depend on the cause of symptom. For example, impulsivity, may be caused by executive (frontal lobe) dysfunction, mood lability, or be associated with attentional deficits. Therefore, a serotonin-reuptake inhibitor could treat frontal impulsivity but is contraindicated for impulsivity due to mood cycling [7]. 

#### 3.1.2. Aggression in Patients with TBI

Aggression includes irritability, anger, agitation, and disinhibition, and represents the most challenging consequences of TBI to manage, interfering with rehabilitation and social support networks. Aggression was found to have a frequency of 5–8% in a sample of military servicemen, who were assessed within the first six months following TBI [43]. Among persons with more severe injuries, the early phase of posttraumatic confusional state is related with a rate of 30–80%, with as many as 20% demonstrating violent behavior. In the late phase of post-injury period following non-penetrating severe TBI, rates of aggression range from 15% to 55% [44,45]. Aggression is influenced by preinjury/post-injury comorbidities [46,47], especially depression and anxiety [48]. 

In animal models, it has been shown that serotonin, dopamine, and adrenaline are neurotransmitters involved in behavioral disorders. In particular, the level of serotonin is negatively correlated with aggressiveness [3], influencing personality changes, agitation, and aggressiveness [49]. Adrenaline and dopamine are involved in attention, memory, and executive functions [50]. 

Neuroimaging studies, assessed by the Buss-Perry Aggression Questionnaire, have revealed that increased diffusivity in the splenium correlated with physical aggression whereas raised diffusivity in the body of the corpus callosum was related to aggressive attitude. Moreover, functional magnetic resonance imaging studies showed a positive association between aggression and increased right hippocampus to midcingulate cortex connectivity, and between the right cerebellar lobule VII region and the default mode network [51,52]. Studies with EEG exclusively showed a positive association between frontal lobe abnormalities and aggression [53,54].

Assessment of the severity and frequency of posttraumatic aggression in individuals with preserved capacity for self-report may be performed by the administration of the Aggression Questionnaire (AQ) [55] and the Personality Assessment Inventory (PAI), investigating posttraumatic aggression and psychiatric symptoms. Among patients with cognitive deficits or BD, the assessment of aggression requires more specific tools. The Modified Overt Aggression Scale could be used for this purpose [56], although when multidimensional neuropsychiatric assessment is needed, the NPI could be administered. 

To treat posttraumatic aggression in an effective way, a multimodal approach involving both non-pharmacologic and pharmacologic interventions, is often necessary. The former includes either replacement or decelerative techniques. Replacement strategies are based on assertiveness training (involving patients who are angry when they fail to get their goals) and several reinforcement scheduling (to reduce the rate of pre-violent behaviors) [57]. Decelerative methods involve contingent observation, social extinction, self-controlled time-out, and contingent restraint [58]. Pharmacologically, frequently aggression in patients with TBI could be treated with carbamazepine, tricyclic antidepressants, trazodone, amantadine; beta-blockers (propranolol, nadolol), valproate, lorazepam, with good tolerability in TBI patients (less frequently with haloperidol, and benzodiazepines). An excessive use of sedative may interfere with rehabilitation project. However, these drugs are limited in terms of use because of the complexity of aggression and the comorbidity with mood disorders or asthma. Therefore, usually, the treatment of aggression is conducted with mood stabilizers [59,60] (see Table 2).

#### 3.1.3. Irritability: An Affect and Mood Disorder

Irritability has been defined as a disproportionate response with unjustified anger fits. Eames [61] suggested that in the early post injury period, irritability is transient because it is arising in response to nearly any stressor or frustration. Instead, irritability in the late postinjury period is marked by regular, cyclic, ego-dystonic outbursts modifying the premorbid affective responding. 

It has been reported an incidence of 29–71% in individuals with severe TBI, being male gender, age range 15–34, unemployment, social isolation and depression the main risk factors [62].

To assess irritability, the self-report Neurobehavioral Symptom Inventory (NSI), a multi-symptom assessment, may be used as screening tool for diagnosing irritability and frustration. However, this tool is limited with regard to its characterization of these specific problems. In patients with limited self-awareness after TBI, the NPI appears to provide functionally important levels of irritability among TBI patients [63].

### 3.2. Affective-Emotional Dysregulation in TBI

Affective-emotional dyscontrol following the brain injury may be related to the disruption at the emotion regulation neural networks [64]. This includes several disorders of affect, such as pathological uncontrollable laughing and crying (PLC), and affective lability, triggered by stimuli that commonly do not cause such emotional feeling state, also known as “pseudobulbar affect” or “emotional incontinence” represent a prototypical form of affective-emotional dyscontrol [45]. These events must cause subjective distress and/or interfere with everyday life activities. The prevalence of PLC during the first year after injury is 5–11% [65] and tends to decrease in the late post-injury period. 

The pathophysiology is still unknown. However, it seems that PLC is the result from the relief of cortical inhibition of upper brainstem centers that incorporate the motor activation patterns involved in laughing and crying. Tateno et al. showed that lesions of the lateral aspect of the left frontal lobe were significantly linked to the presence of PLC [45]. In particular, the prefrontal cortex modulates the emotional motor and autonomic responses of sensory and limbic information involved in emotional expression. Therefore, lesions of prefrontal control of hypothalamic, pontine, and medullary centers that mediate these responses lead to PLC. Thus, a correlation between PLC and the dorsolateral/anterior frontal cortices, internal capsule, prefrontal cortex, amygdala, hippocampus, basal ganglia, and thalamus has been show, leading to a chronic condition (cortico-limbic–subcortico-thalamic–ponto-cerebellar network), beyond the cortical inhibition of upper brainstem centers involved in the integration of motor activation patterns [66]. It has been shown that PLC is associated with psychiatric disorders, especially anxiety and mood disorders (depression in particular) and personality changes [45,67]. The quantitative scale that could measure and quantify the severity of PLC is the Pathological Laughter and Crying Scale [68].

The first-line treatment of PLC is pharmacotherapy with serotonin reuptake inhibitors (SSRI) and/or selective noradrenaline reuptake inhibitor (NaRI) active antidepressants [45,68,69,70]. Moreover, in a recent case report, a patient with TBI and PLC was treated with dextromethorphan/Quinidine (DM/Q 20 mg/10 mg, Nuedexta) showing an improvement in PLC. Dextromethorphan is a non-competitive antagonist of the NMDA receptors and a sigma receptor agonist, having a neuroprotective effect on injured brain [71]. 

On the other hand, quinidine is a potent cytochrome P450 2 D6 inhibitor and reduces the metabolism of DM [72]. 

Affective lability (also known as emotional lability) refers to a tendency to be easily overcome with intense emotions in response to personally, socially meaningful stimuli or events that ordinarily would induce modest emotional responses [58]. It manifests as brief, non-stereotyped episodes of congruent emotional expression and experience that are not discretely paroxysmal, of variable intensity, and partially amenable to voluntary control or interruption by external events. Affective lability is observed in association with a broad range of psychiatric and medical conditions [73] with a prevalence ranging from 33% to 46% in the early post-injury period, and to 14–62% in the late period [74]. The Affective Lability Scale or the CNS-LS is a self-report scale to evaluate affective lability [75].

### 3.3. Empathy and Alexithymia Following TBI

The presence of alexithymic behaviours in people who had an acquired brain injury as defined as ‘organic alexithymia’ [76,77] and may considered as a deficit in affective-cognitive handling [78]. Respectively, people with alexithymia reveal impairment in emotional orientation, consciousness, and communication of their emotions [79]. The prevalence of organic alexithymia has been estimated at approximately 15% in the acquired brain injury population. Consequences of alexithymia are verbal explosions, maladaptive coping, impulsivity, and aggression with a lack of consideration for other’s perspectives [79,80]. The limit in awareness of emotion impedes the efficiently application of coping strategies inhibiting an adequately expression and communication of emotions, leading to increased risk of suicidal ideation [38]. 

Higgins and Endler [81] identified the main strategies for coping with psychological distress: (1) task-oriented coping, (2) social-emotional coping, and (3) avoidance-oriented coping [82]. In contrast, social-emotional coping has proven to be a significant risk factor in the development of anxiety and depression and has been linked to poor quality of life. Avoidance-oriented coping seems to create the highest risk of psychological distress. On the contrary, those who apply task-oriented coping over a long period are more likely to report a better quality of life. Recent research on personality traits suggests that alexithymia may influence the coping styles following severe TBI [83]. Krpan et al. [84] found that individuals with executive dysfunction and difficulty identifying feelings showed a greater use of avoidance coping and loss of empathy [50], in agreement with other studies that have demonstrated alexithymia to be more associated with avoidant than task oriented. 

Alexithymia has been associated with dysfunction in the right hemisphere cortex, the corpus callosum, anterior cingulate cortex, front striatal networks, and amygdala [77,85]. A relation between brain–injury and high alexithymia and psychiatric disorders has been shown [45]. [86,87]. This has been confirmed in a number of studies using the 20-item Toronto Alexithymia Scale (TAS-20) [86,87]. The scale is a multi–dimensional self–report inventory constructed to assess the three dimensions of alexithymic traits: (1) difficulty identifying one’s feelings (DIF); (2) difficulty describing one’s feelings (DDF); and (3) externally orientated thinking (EOT) [79]. The cut off for diagnosis of alexithymia is ≥61 [88]. In a recent meta-analysis on the estimation of the prevalence of alexithymia in patients with acquired brain injury, [89] it was found that the magnitude of the effect size of global alexithymia was approximately 25–30%, larger with respect to others meta-analysis on the effect of cognitive ability. Additionally, moderate–to–large effect sizes were reliable through the TAS–20 subscales, i.e., DIF, DDF, and EOT. Moreover, comparing the impact brain lesions on the emotion disorders between patients with TBI with patients with stroke or tumors (i.e., non–TBI), the authors found that patients with a TBI evidenced statistically significantly greater degrees of alexithymia, respect to non–TBI patients. Therefore, the lesions in TBI occur in the brain area responsible for identifying emotions, describing emotions, and externally oriented thinking. Thus, it is important to assess TBI patients for emotional deficits [88].

The risks of alexithymia after TBI are important to identify. It has been shown that one risk is represented by gender. A meta-analysis in the general population reported the same discrepancy, explained by the “normative male hypothesis”, as males during childhood are dejected to feel or exhibit or transfer any kind of emotions [89]. A recent study on gender differences in terms of alexithymia in TBI patients did not found dissimilarities between women and men in the TAS score, but the authors found a higher proportion of alexithymia in TBI patients respect to uninjured controls [90]. An association between alexithymia and empathy in the TBI population has been showed [91,92]. Moreover, there has not been shown to be a sex difference in empathy between men and women with TBI. However, 44% of women with TBI fell below emotional empathy norms than men with TBI [93]. Another study showed that men with TBI were more impaired in empathy than women [91]. An inverse relationship between alexithymia and empathy in clinical and nonclinical populations has been shown [94,95,96].

As a construct, empathy can be divided into emotional and cognitive components [97]. The emotional problems exhibited by many TBI people often leads to relationships failure. Emotional empathy refers to the capacity to feel the emotions of others and has been associated with the insula and inferior frontal gyrus [98]. On the other hand, cognitive empathy refers to the aptitude to understand another’s viewpoint, and it has been linked to the ventromedial prefrontal cortex [97,98,99]. Wood et al. [87] suggested the evaluation of emotional empathy by the Balanced Emotional Empathy Scale (BEES). 

## 4. Depression, Anxiety and Post-Traumatic Stress Disorder (PTSD)

Although during their first year after injury, approximately 14–61% of patients with TBI may experience depressed mood, the prevalence of depression, as well as anxiety, reduced progressively over five years, with rates similar to general population. The most common diagnosis was Major Depressive Disorder and Anxiety Disorder NOS (usually PTSD or generalized anxiety disorder) [100]. Recently, the American Psychological Association (APA) has developed a website with the overall assessments and complete guideline treatment of PTSD for a better diagnosis and treatment of the disorder [101].

The events that cause TBI might be severely emotionally traumatic. Even a psychiatric disorder before TBI was a strong predictor. However, most of the studies showed that mood and anxiety disorders were developed de novo [17,90,102]. Symptoms of PTSD are re-experiencing symptoms, escaping, and impairment of arousal, cognition, and mood. Risk factors of PTSD are TBI, less education, being black, and the kid of injury. It has been shown that mild TBI raised predicted PTSD symptoms by a factor of 1.23, while moderate or severe TBI amplified predicted symptoms by a factor of 1.71 [19,103]. Moreover, the risk of developing panic disorder (odds ratio = 2.01), social phobia (odds ratio = 2.07), and agoraphobia (odds ratio = 1.94) in mild TBI was twice doubling [104]. Anxiety is often related to injury to the mesial temporal lobe and can present as generalized anxiety disorder (GAD), panic disorder, obsessive–compulsive disorder (OCD), posttraumatic stress disorder (PTSD), or different phobias (as per DSM-V) [105,106]. Nonetheless, there is overlap between the cognitive deficits of PTSD and those due to TBI itself, with a rate of 3–23% in more severely affected patients. PTSD is a complex disorder that develops after a traumatic situation [107,108]. This risk is very greater in severe TBI patients [109]. PTSD and TBI are usually linked with neuropsychological impairments of executive functions, attention and working memory. In TBI PTSD patient populations, there has been hypothesized to be an effect on the amygdala and an endocrine malfunction that may underlie a vulnerability to the development of PTSD following subsequent injury and/or stressor exposure [110]. Actually, a clear neurobiological basis for understanding the complexity of PTSD is still unknown. Three theories of pathophysiology are valid: one depends on the negative effect of TBI on neural circuits regulating fear responses [111]. 

In TBI patients, as well as in the general population, functional neuroimaging studies have shown impaired brain activity in the dorsolateral and ventrolateral prefrontal cortices as well as anterior cingulate, which are known to subtend emotional regulation self-control and [112]. The second is based on the consequently cognitive impairment after TBI that could negatively influenced the coping capacity using adaptive cognitive strategies [113]. Lastly, the inflammatory theory after TBI that might contribute to aggravated mental health outcomes [114]. Also, because of similar symptoms of affective and cognitive deficits in depressive patients without TBI respect to TBI patients, a clear distinction needs to be made determining if the depressed mood is situational and/or reactive to the trauma, primary mood disorder, or secondary to either a structural lesion or a specific medical condition [115]. Patients with major depression or depressive disorder not otherwise specified (NOS) might be treated with antidepressants, such as selective serotonin reuptake inhibitors, sertraline, or citalopram, as a recommended first-line treatment [116]. Even if less studied, selective serotonin-epinephrine reuptake inhibitors have been recommended [117]. On the other hand, due to the effect of lowering the seizure threshold in patients with moderate–severe TBI, tricyclic antidepressants are not suggested [118,119]. 

Nonpharmacological treatment with psychotherapy has to be proposed as an alternative or adjunctive treatment [7]. Telerehabilitation is a valid and reproducible therapy as effective as in-person therapy [120]. 

## 5. Suicide Attempts and Suicide Ideation

Patients with TBI have a rate of suicide attempts and suicide ideation (SI) as well as suicide higher than the general population, even after checking for psychiatric problems. The suicide relative risk in TBI severe was 3–4 times higher than the general population whereas SI was found in 21–22% in individuals with TBI, and it has been associated to a higher risk of suicide attempts. Suicide attempts indeed could reach 18% [121]. Findings on the causes of post-injury SI are heavily limited. Some clinicians have suggested that the dysregulation of serotonergic and noradrenergic neural pathways that results from TBI may be implicated in suicidality. No evidence, however, is available to support this hypothesis. Lately, suicidal behavior has been related with decreased brain-derived neurotrophic factor functioning.

A relationship between neuropsychological impairment and SI has been argued as well. Executive dysfunction functioning has been associated with suicidality in non-TBI samples [122]. Patients with depression and a history of attempted suicide or ongoing SI tend to perform significantly worse on measures of executive functioning, problem solving and mental flexibility, as compared to depressed peoples with no history of suicide attempt/ideation. Depression and history of emotional/psychiatric disturbance are associated with completed and attempted suicide in severe TBI patients. Indeed, mood, personality, and substance abuse disorders are identified as significant risk factors for suicidal behavior in the general population. As would be expected, depression has been identified as the most significant correlate of post-TBI SI, although no data exist demonstrating the amount of variance in SI explained by depression. It has been shown that patients with TBI and depression were six times more likely to report SI compared with people with minimal or borderline depression [123]. Few systematic investigations have explored the relationship between postinjury psychosocial functioning and suicide ideation (SI). Among the others, employment seems to play a role in SI, whereas adverse life events postinjury, social isolation, and relationship breakdown have been shown not to be related to the presence of postinjury suicide plans. 

## 6. Discussion and Clinical Advice

TBI represents a main cause of death and disability worldwide. Indeed, many severe TBI survivors present with long-term physical, neuropsychiatric, and cognitive disorders with a need of long-term rehabilitation [124]. Notably, some TBI patients may recover motor function, but neuropsychological and behavioral problems (the so-called “invisible disability”) persist months or years after the brain injury [125]. Given that NPS may affect every aspect of an individual’s life, an adequate assessment of TBI must include the investigation of cognitive (including memory and executive function abnormalities) and neuropsychiatric problems (such as anxiety, depression, agitation, and substance abuse) [125]. Nonetheless, only a few studies have dealt with the effect of NPS on patients’ and caregivers’ social and occupational reintegration [126].

Personality changes, intolerance to frustration, anxiety, depressive mood, anger, or aggression negatively affects both patient and family quality of life [127]. For a better management of this disabling problem, the first recommendation concerns the need for all healthcare professionals to use common definitions and nomenclature for neuropsychiatric and behavioral abnormalities. When consensus lacks, the disorders should be classified in “disruptive primary behaviors, affective disorders, anxiety-psychosis and suicide attempts/suicide”. Older and more recent psychiatric guidelines, i.e., ICD-10 and DSM-V, report limits in the description and classification of TBI-related disorders. However, the psychiatric aspects of TBI have expanded and improved through the last several editions of the American Psychiatric Association’s Diagnostic and Statistical Manual of Mental Disorders (DSM) [34,127], encouraging clinicians to consider risk factors that may modify TBI outcomes and treatment [128,129,130]. 

Secondly, the use of specific scales tailored to the patients with TBI is fundamental for an accurate assessment of these frail individuals. This is why we have pointed out the most commonly used scales to investigate these still often disregarded post-TBI problems (Table 3).

**Table 3 biomedicines-11-01449-t003:** Behavior and personality assessment in TBI.

Cognitive Domain	Authors	Test	Short Description of Test	Time to Administer
Personality Emotional Conditions Social cognition	Schroeder et al. (2019) [131]	Minnesota Multiphasic Personality Inventory-2 (MMPI-2)	The Minnesota Multiphasic Personality Inventory (MMPI) is a psychological instrument used to identify signs of psychopathology. The MMPI-2 contains around 567 items.	60–90 min
	Schroeder et al. (2019) [131]	Personality Assessment Inventory (PAI)	The Personality Assessment Inventory (PAI) is a self-report measure of personality and psychopathology consisting of four Validity, eleven clinical, five Treatment Consideration, and two Interpersonal scales, as well as conceptually derived subscales and risk indexes	Requires 50–60 min
	Westerhof-Evers et al. (2014) [132]	The Awareness of Social Inference Test (TASIT)	The The Awareness of Social Inference Test (TASIT) is used to evaluate the patients’ ability to understand emotional states, thoughts, intentions and conversational meaning in everyday exchanges.	1 h
	Gorgoraptis et al. (2019) [133]	Beck Depression Inventory–Second Edition (BDI-II)	The Beck Depression Inventory (BDI-II) is a 21-item self-report tool assessing the severity of depression symptoms.	10–15 min
	Bertisch et al. (2013) [134]	Beck Anxiety Inventory (BAI)	The Beck Anxiety Inventory (BAI) is a brief, self report assessment for anxiety.	5–10 min
	Quaranta et al. (2008) [135]	Poststroke Depression Rating Scale (PSDRS)	The Poststroke Depression Rating Scale (PSDRS) is a clinical tool specifically devised to assess depression after stroke.	10 min
	Benaim et al. (2004) [136]	Aphasic Depression Rating Scale (ADRS)	The Aphasic Depression Rating Scale (ADRS) was developed to detect and measure depression in aphasic patients during the subacute stage of stroke.	8 min
	Kieffer-Kristensen and Teasdale (2011) [137]	Symptom checklist -90 R questions (SCL-90 R)	The Symptom checklist -90 R questions (SCL-90-R) is a largely-used questionnaire for self-report of psychological distress and multiple aspects of psychopathology, as part of the evaluation of chronic pain patients	It takes 15–20 min to administer
	Diaz et al. (2012) [17]	Brief psychiatric rating scale (BPRS)	The Brief Psychiatric Rating Scale (BPRS) is one of several tools used to evaluate patients with schizophrenia and related psychotic disorders. They measure it to track changes in symptoms over time.	15 min
	Malec et al. (2018) [138]	Neuropsychiatric Inventory (NPI)	The Neuropsychiatric Inventory (NPI) is a clinical interview with a family member or friend who knows the patient well and can evaluate 12 behavioral areas commonly affected in patients with dementia.	Taking roughly 20 min

The assessment of personality and psychopathology following TBI is an important aspect of clinical care and rehabilitation. Although the most overt symptoms of moderate-to-severe TBI are cognitive and physical disorders, personality and psychiatric disturbances are important as they persist even after cognitive and physical symptoms resolve [139]. The use of a comprehensive and multi-modal approach is then fundamental in the evaluation of behavioral symptoms and personality disorders in severe TBI patients. Among the others, the Personality Assessment Inventory (PAI) is a self-report measure of personality and psychopathology that has demonstrated good convergent validity for a small subset of items among TBI persons [140]. 

The Neuropsychiatric Inventory (NPI) is a clinical scale that evaluates psychiatric symptoms in patients with neurological disorders and severe traumatic brain injury (TBI) patients to explore neurobehavioral impairments. 

Third, to reduce and better manage the psychological and behavioral consequences of TBI, clinicians should be aware of the available pharmacological and psychotherapeutic approaches.

Two important reviews have concluded that there is limited evidence for effective pharmacological treatment of aggression after TBI, lacking in available treatment guidelines [141,142]. However, both reviews showed the potential utility of beta blockers to decrease the intensity of agitated episodes and the frequency of assault attempts, although there is often a need for high doses to achieve the clinical outcomes with risk of hypotension and bradycardia. The use of SSRI antidepressants could be of help in the treatment of impulsive and episodic aggression [143]. Although neuroleptics, such as chlorpromazine and haloperidol, can cause severe sedation, they can be used to control aggression. Atypical antipsychotics, such as quetiapine, are believed to be less harmful [143].

Cognitive behavioral therapy (CBT) has been successfully employed to manage behavioral and neuropsychiatric symptoms, such as impulsive aggression in a variety of settings [144]. In clinical settings, CBT interventions have also demonstrated efficacy in treating intermittent explosive disorder, leading to a reduction in aggressive acts, hostile thinking, and associated depressive symptoms, by adaptive self-regulating behaviors or simply avoidance behaviors that will minimize the risk of violence [145]. Some authors [58,146] have presented guidelines on how to apply CBT methods in the management of aggression after TBI but, so far, no clinical research has been published to examine the effectiveness of CBT in the treatment of these conditions. In contrast, there is a considerable body of literature on behavior management techniques, especially in the clinical management of irritability, disinhibition, and impulsive aggression [58]. By now, the psychological methods that have been shown useful in a variety of rehabilitation and community settings, are those that utilize operant learning theory, often combined with cognitive methods such as verbal mediation [58]. Another efficacious therapy is the replacement strategy, which includes assertiveness training (intended for patients who become angry when they fail to get their needs met) and differential reinforcement scheduling (to decrease the rate of pre violent behaviors) [147]. Effective treatment of behavioral symptoms and posttraumatic aggression focuses on a multidisciplinary approach involving both non-pharmacologic and pharmacologic interventions. Patients diagnosed with disorders who present with mood lability might be treated with mood stabilizers over 95% of the time. Because of their efficacy in the treatment of agitation and mood lability, mood stabilizers might also be effective in the treatment of personality change or cognitive dysfunction, such as impulsivity and distractibility [148].

## 7. Conclusions

Behavioral dyscontrol, neuropsychiatric and emotional symptoms are relatively common and disabling consequences of TBI, especially in the case of severe injury. To better understand the phenomenology and epidemiology and effectively manage these problems, a multidisciplinary approach is required. Valid and reliable symptom-specific and multidimensional neuropsychological metrics are useful to assess these disorders. Emotional, neurocognitive, and BD frequently co-occur with other posttraumatic neuropsychiatric disturbances, and this should be considered when dealing with these frail patients and their caregivers. Moreover, a symptom-specific treatment characterized by a combination of non-pharmacologic, i.e., psychological, behavioral, and environmental, and pharmacologic approaches is usually required. When properly administered, these interventions may provide severe TBI patients and their families with substantial relief from posttraumatic emotional and behavioral impairments.

## Figures and Tables

**Table 1 biomedicines-11-01449-t001:** The wide spectrum of “Frontal Personality” and other syndromes.

Clinical Syndrome	Brain Lesions	Personality Changes, Prevailing Signs
Prefrontal syndrome	extensive unilateral-bilateral lesions of the frontal lobe	uninhibited, attitudes of swagger, variable mood, euphoric, with lack of worries or with excessive worries, with self-centeredness, obstinacy apathy, indifference, distractibility, neglect in clothing, intolerance to frustrations, eating disorders, drowsiness, lethargy
Pseudo-psychopathic	Lesions of the fronto-orbital cortex	euphoria, restlessness, sexual disinhibition, childishness, inappropriate social behaviour, lack of interest
Psychomotor slowdown	Lesions of the medial frontal cortex: dorsal region (cingulate gyrus and supplementary motor area),	psychomotor slowdown (the patient only answers when he is asked direct questions) abulia
Cognitive-behavior syndrome	Ventro-medial frontal lesions (limbic cortex and diencephalic structures)	severe memory impairment, socially inappropriate behavior, disorientation, changes in consciousness, apathy and uncontrolled aggression.
Cognitive impairment	dorsolateral lesions (the orbital and medial frontal areas)	neurocognitive deficits spare many aspects of mental functioning.
Cerebellar cognitive affective syndrome	Cerebellum	emotional dyscontrol, aggression, Cognitive dysfunction (executive, visuospatial, and language function

**Table 2 biomedicines-11-01449-t002:** Aggression therapy with only Expert opinion and guidelines [59,60,61].

Class of Therapy	Indication	Mechanism of Action	Dosage	Expert Opinion
Beta-blockers (propanolol)	Agitation management	/	60−80 mg per day, initially 20 mg/day, then gradually increased up to a maximum of 420 mg per day (60 mg raises every 3 days) and a follow-up of 8-weeks (if no effects after 8 weeks, stop it)	Propranolol is the Gold Standard when a TBI patient suffers from both high-blood pressure and agitation. Recommended to perform an electrocardiogram before beginning the treatment
Antiepileptics	Mood-regulating	Act on neurotransmitters: glutamatergic GABAergic agents limbic system	Oxcarbazepine 1200 mg/day, (max 2400 mg over a 10-week period) carbamazepine 400–800 mg/day for 8-weeks lamotrigine 50 mg twice a day	Gold Standard for agitation or aggressiveness, after TBI, when associated with epilepsy or bipolar disorders
Antidepressants	Agitation and aggressiveness	Act on neurotransmitters involved in behavioral disorders: serotonin, dopamine and adrenaline.	Sertraline 100 mg/day (max 200 mg/day, first line) Amitriptyline 25 mg/day (second-line treatment)	Gold standard when agitation or aggressiveness crisis is associated with depression
Neuroleptics	Agitation crisis	Sedative	Olanzapine, clozapine, quetiapine 25–300 mg/day ziprasidone 20–80 mg/day	Neuroleptics can be used in the treatment of agitation associated with delirious symptoms
Benzodiazepines	Agitation crisis	/	Clorazepate dipotassium	Indicated when anxiety is the predominant symptom (symptomatic prescription)

## Data Availability

Not applicable.

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
