# Peer review of "Behavioral and Psychiatric Symptoms in Patients with Severe Traumatic Brain Injury: A Comprehensive Overview"

_biomedicines, 2023, doi:10.3390/biomedicines11051449_

Round 1
Reviewer 1 Report
Many thanks for this well written manuscript which provides a comprehensive overview of the behavioural and psychiatric symptoms after severe TBI.
I would suggest to include in your manuscript the following considerations:
1) In section 2 please add that "delirium (Ganau M, Lavinio A, Prisco L. Delirium and agitation in traumatic brain injury patients: an update on pathological hypotheses and treatment options. Minerva Anestesiol. 2018;84(5):632-640. doi: 10.23736/S0375-9393.18.12294-2.) and status epilepticus (Prisco L, Ganau M, Aurangzeb S, Moswela O, Hallett C, Raby S, Fitzgibbon K, Kearns C, Sen A. A pragmatic approach to intravenous anaesthetics and electroencephalographic endpoints for the treatment of refractory and super-refractory status epilepticus in critical care. Seizure. 2020;75:153-164. doi: 10.1016/j.seizure.2019.09.011.) in neurointensive care unit are very well know risk factors for developing behavioural disorders at long term follow up"
2) The use of EEG in NICU adopted by Prisco et al. corroborates the statement in lines 106-108. In section 3 I believe it would be appropriate to mention the body of literature on CTE (McKee AC, Stein TD, Huber BR, Crary JF, Bieniek K, Dickson D, Alvarez VE, Cherry JD, Farrell K, Butler M, Uretsky M, Abdolmohammadi B, Alosco ML, Tripodis Y, Mez J, Daneshvar DH. Chronic traumatic encephalopathy (CTE): criteria for neuropathological diagnosis and relationship to repetitive head impacts. Acta Neuropathol. 2023;145(4):371-394. doi: 10.1007/s00401-023-02540-w.)
3) In section 4 I would suggest referencing to the tools currently available to optimise diagnosis and management of PTSD. Our team relies heavily on APA guidelines on PTSD and it would be good to list those tools in your manuscript by inserting the hyperlink in the text (https://www.apa.org/ptsd-guideline/assessment) because they are not among those listed in section 5.
Other than those recommendations for minor revision, as far as I am concerned I believe that this manuscript would be ready for acceptance.
I congratulate again the authors for their passion toward the management of TBI patients and contribution to the existing literature on this challenging area of multidisciplinary clinical practice.
Author Response
Many thanks for this well written manuscript which provides a comprehensive overview of the behavioural and psychiatric symptoms after severe TBI.
I would suggest to include in your manuscript the following considerations:
1) In section 2 please add that "delirium (Ganau M, Lavinio A, Prisco L. Delirium and agitation in traumatic brain injury patients: an update on pathological hypotheses and treatment options. Minerva Anestesiol. 2018;84(5):632-640. doi: 10.23736/S0375-9393.18.12294-2.) and status epilepticus (Prisco L, Ganau M, Aurangzeb S, Moswela O, Hallett C, Raby S, Fitzgibbon K, Kearns C, Sen A. A pragmatic approach to intravenous anaesthetics and electroencephalographic endpoints for the treatment of refractory and super-refractory status epilepticus in critical care. Seizure. 2020;75:153-164. doi: 10.1016/j.seizure.2019.09.011.) in neurointensive care unit are very well known risk factors for developing behavioural disorders at long term follow up"
Thank you for your suggestion. We added the sentence and references in the text.
2) The use of EEG in NICU adopted by Prisco et al. corroborates the statement in lines 106-108. In section 3 I believe it would be appropriate to mention the body of literature on CTE (McKee AC, Stein TD, Huber BR, Crary JF, Bieniek K, Dickson D, Alvarez VE, Cherry JD, Farrell K, Butler M, Uretsky M, Abdolmohammadi B, Alosco ML, Tripodis Y, Mez J, Daneshvar DH. Chronic traumatic encephalopathy (CTE): criteria for neuropathological diagnosis and relationship to repetitive head impacts. Acta Neuropathol. 2023;145(4):371-394. doi: 10.1007/s00401-023-02540-w.)
Thank you for your important and interesting suggestion. We added few sentences on the pathological characteristics of CTE in section 3, as suggested.
“Pathologically, chronic traumatic encephalopathy (CTE) is characterized by an accumulation of neuronal phosphorylated tau (p-tau) in the perivascular regions and p-tau fibrils as in Alzheimer’s disease, or tauopathy [23].”
3) In section 4 I would suggest referencing to the tools currently available to optimise diagnosis and management of PTSD. Our team relies heavily on APA guidelines on PTSD and it would be good to list those tools in your manuscript by inserting the hyperlink in the text (https://www.apa.org/ptsd-guideline/assessment) because they are not among those listed in section 5.
We really appreciate your recommendations. We improve the text content reporting the suggested research. We added the following sentence on the text and the related fundamental reference and website link.
“Recently, the American Psychological Association (APA) has developed a website the overall assessments and complete guideline treatment of PTSD for a better diagnosis and treatment of the disorder [101].”
Other than those recommendations for minor revision, as far as I am concerned, I believe that this manuscript would be ready for acceptance.
I congratulate again the authors for their passion toward the management of TBI patients and contribution to the existing literature on this challenging area of multidisciplinary clinical practice.
Thank you very much for your positively relevant comments on our paper.

Reviewer 2 Report
This comprehensive review addresses the behavioral and psychiatric symptoms in patients with severe traumatic brain injury.
Line 12: Change to "Traumatic brain injury (TBI) is defined..."
Line 16: Change "showed a prevalence" to "shown prevalence"
Line 18: Change "...is carried out..." to "...occurs..."
Line 19: Change to "...1 to 2 years the condition becomes stable." Also change "...risk factors of poor..." to "...risk factors for poor..."
Line 23: Change "Aim of..." to "The aim of..." As an overview it isn't really an investigation. Please rephrase this line.
Line 31: Change "...due to the high..." to "...due to its high..."
Line 124: Change "propency" to "propensity"
Line 127: delete the full stop after [29].
Line 151: Change to "consequences"
Line 193: Delete "Rietdijk et al. (2020)"
Lines 212 - 213: By "frequency" do you mean the number of incidents or individuals who experienced aggression? please make this sentence clearer.
Line 274: Please check the in-text referencing convention, but I'm don't think you need to include the year
Line 702: This reference is incorrectly formatted.
Author Response
This comprehensive review addresses the behavioral and psychiatric symptoms in patients with severe traumatic brain injury.
Line 12: Change to "Traumatic brain injury (TBI) is defined..."
Line 16: Change "showed a prevalence" to "shown prevalence"
Line 18: Change "...is carried out..." to "...occurs..."
Line 19: Change to "...1 to 2 years the condition becomes stable." Also change "...risk factors of poor..." to "...risk factors for poor..."
Line 23: Change "Aim of..." to "The aim of..." As an overview it isn't really an investigation. Please rephrase this line.
Line 31: Change "...due to the high..." to "...due to its high..."
Line 124: Change "propency" to "propensity"
Line 127: delete the full stop after [29].
Line 151: Change to "consequences"
Line 193: Delete "Rietdijk et al. (2020)"
Lines 212 - 213: By "frequency" do you mean the number of incidents or individuals who experienced aggression? please make this sentence clearer.
Line 274: Please check the in-text referencing convention, but I'm don't think you need to include the year
Line 702: This reference is incorrectly formatted.
Thank you very much for your comments. We reported all the suggested adjustment of the text.
